# Heart Rate Variability and Interoception in Periodic Limb Movements in Sleep: Interference with Psychiatric Disorders?

**DOI:** 10.3390/jcm13206129

**Published:** 2024-10-14

**Authors:** Marta A. Małkiewicz, Krzysztof S. Malinowski, Małgorzata Grzywińska, Eemil Partinen, Markku Partinen, Jan Pyrzowski, Magdalena Wszędybył-Winklewska

**Affiliations:** 1Applied Cognitive Neuroscience Lab, Department of Neurophysiology, Neuropsychology and Neuroinformatics, Medical University of Gdansk, 80-210 Gdansk, Poland; 2Department of Neurophysiology, Neuropsychology and Neuroinformatics, Medical University of Gdansk, 80-210 Gdansk, Poland; krzysztof.malinowski@gumed.edu.pl (K.S.M.); magdalena.wszedybyl-winklewska@gumed.edu.pl (M.W.-W.); 3Neuroinformatics and Artificial Intelligence Lab, Department of Neurophysiology, Neuropsychology and Neuroinformatics, Medical University of Gdansk, 80-210 Gdansk, Poland; malgorzata.grzywinska@gumed.edu.pl; 4Helsinki Sleep Clinic, Terveystalo Healthcare, 00380 Helsinki, Finland; eemil.partinen@gmail.com (E.P.); markpart@me.com (M.P.); 5Department of Neurology, Helsinki University Central Hospital, 00260 Helsinki, Finland; 6Department of Neurosciences, Clinicum, University of Helsinki, 00100 Helsinki, Finland; 7Department of Emergency Medicine, Medical University of Gdansk, 80-210 Gdansk, Poland; jan.pyrzowski@gumed.edu.pl

**Keywords:** periodic limb movements in sleep, heart rate variability, autonomic nervous system, cardiovascular interoception, psychiatric disorders

## Abstract

Periodic limb movements in sleep (PLMS) are a prevalent disorder characterized by rhythmic, involuntary movements of the lower limbs, such as dorsiflexion of the ankle and extension of the big toe, occurring in periodic intervals during sleep. These movements are often linked to disrupted autonomic nervous system (ANS) activity and altered interoception. Interoception involves perceiving internal bodily states, like heartbeat, breathing, hunger, and temperature, and plays a crucial role in maintaining homeostasis and the mind–body connection. This review explores the complex relationships between PLMS, heart rate variability (HRV), ANS dysregulation, and their impact on psychiatric disorders. By synthesizing the existing literature, it provides insights into how ANS dysregulation and altered interoceptive processes, alongside PLMS, contribute to psychiatric conditions. The review highlights the potential for integrated diagnostic and therapeutic approaches and presents a cause-and-effect model illustrating the mutual influence of psychiatric disorders, ANS dysregulation, PLMS, and interoception.

## 1. Introduction

Periodic limb movements in sleep (PLMS) are involuntary, stereotyped movements of the lower limbs that occur in a periodic interval pattern during sleep, typically involving dorsiflexion of the ankle, extension of the big toe, and occasional flexion of the knee and hip. PLMS are present in 80% of patients with restless leg syndrome (RLS) but also in particular sleep disorders, psychiatric diseases, and neurological pathologies. Emerging evidence suggests a potential link between PLMS and psychiatric disorders, particularly anxiety and depression.

While PLMS primarily pertains to sleep disturbances, its relationship with psychiatric disorders is multifaceted. The role of autonomic nervous system (ANS) impairment in PLMS has been widely investigated, particularly the involvement of the sympathetic nervous system. This review aims to explore the relationship between ANS activity in PLMS and mental diseases by examining heart rate variability (HRV). HRV reflects the intracardiac activity of the ANS and, thus, the complexity of the regulatory mechanisms of the cardiovascular system. HRV may, therefore, indirectly serve as a marker of ANS health. Disease processes, aging, lifestyle, and external environmental and neuropsychological conditions may lead to disturbances in the autonomic regulation of the heart rate (HR) and thus influence HRV [1,2]. Smoking cigarettes or excessive alcohol consumption reduces HRV, while an active lifestyle, regular physical exercise, and practicing relaxation methods, including meditation, increase HRV parameters [3]. The findings highlight the importance of understanding the autonomic control of HR and its implications in the context of PLMS and mental psychiatric disorders. While some studies have reported ANS dysfunction in PLMS and psychiatric conditions based on HRV analysis, the underlying pathogenesis and the precise relationship between PLMS, mental diseases, and ANS activity remain to be fully elucidated.

Interoception, which involves the perception of internal bodily states, adds another layer of complexity to our understanding of these relationships. Interoception entails the recognition of feelings related to body functions per the cardiovascular system (such as heartbeat), breathing, hunger, thirst, temperature, and other visceral experiences. It helps maintain the stability of the internal environment (homeostasis) but also plays a significant role in the mind–body connection by managing emotions and making decisions based on the internal state [4].

By synthesizing the existing literature, this review provides insights into how ANS dysregulation, altered interoceptive processes, and PLMS collectively contribute to psychiatric disorders, emphasizing the potential for integrated diagnostic and therapeutic approaches. For a better thematization of the discussed topic, we formulated a cause-and-effect sequence, which, in the form of a closed circle, covers the mutual influence of psychiatric disorders, ANS dysregulation, PLMS, and interoception (Figure 1).

## 2. Psychiatric Disorders and ANS Dysregulation (HRV)

The timing of adjacent cardiac cycles is variable both at rest and in states of arousal. Intracardiac ANS fibers constitute the efferent pathway of the cardiac reflexes involved in the complex regulatory mechanisms of heart function. HRV is the result of biological oscillators that generate repetitive rhythms. The frequency and amplitude of these oscillations are a valuable indicator of the body’s functional state and ability to adapt to changing environmental conditions [5]. The basic variable analyzed is the NN (normal to normal) interval [i.e., standardized R-wave peak to R-wave peak (RR) intervals, verified for the correctness of their determination]. On their basis, parameters regarding the average length of the RR interval (i.e., the average duration of the cardiac cycle), as well as the total and short- and long-term variability, are determined [6]. Time domain analysis is based on computational methods of mathematical statistics. This is the oldest method and, at the same time, characterized by the least complexity. The second class of methods is based on analysis of the frequency domain using a non-parametric method, the discrete Fourier “transform”, or a parametric method based on autoregression. Harmonic analysis involves separating a complex rhythm into components that are harmonic curves with different frequencies and amplitudes. Spectral analysis calculates the contribution of individual harmonic frequencies to the total irregularity of the recording and presents them in the form of a signal power density spectrum [7]. The low-frequency (LF) component of HRV reflects the combined impact of both the sympathetic and parasympathetic nervous systems, whereas the high-frequency (HF) component is chiefly under the control of parasympathetic signals transmitted to the heart via the vagal nerve.

HRV reflects the variation in heartbeat intervals, regulated by the ANS, and is influenced by physiological factors. Studies show that while a healthy heart exhibits complex nonlinear dynamics and long-range correlations, the reduction of HRV in some medical conditions leads to a reduction in its complexity by disrupting these patterns [8,9].The inclusion of nonlinear HRV measures that evaluate data structure and organization shows promise for improving HRV interpretation. Assessing chaotic and nonlinear dynamics in HRV can be used for understanding the complex behavior of the cardiovascular system, particularly its adaptive responses to physiological and pathological conditions. While traditional time- and frequency-domain measures of HRV are useful, they often fail to capture the intricate dynamics of heart rate regulation. Nonlinear methods, inspired by chaos theory and fractal mathematics, can provide deeper insights into the complex variability of heart rate signals [10].

Cardiac regulation via the ANS involves a complex interplay between higher brain regions and brainstem centers responsible for autonomic functions. The ANS is a multifaceted neural network that orchestrates the involuntary physiological processes essential for maintaining homeostasis. Its intricate regulation involves a dynamic interplay between higher brain regions, brainstem centers, and neural circuits. The rostral ventrolateral region of the medulla oblongata (RVLM) plays an important role in generating and regulating sympathetic activity in the cardiovascular system [11]. Neurons in the RVLM have the ability to generate rhythmic action potentials and determine tonic central sympathetic activity. The RVLM receives information from peripheral receptors, including pulmonary baroreceptors and mechanoreceptors, arterial chemoreceptors, inspiratory neurons, central chemodetectors, and ergoreceptors [12].

Moreover, projections from higher structures of the nervous system reach the RVLM area. Influences coming from the prefrontal cortex, amygdala, hypothalamus, and hippocampus are involved in emotional and defensive reactions [13]. The cerebral cortex, particularly areas involved in cognitive, emotional, and behavioral control, plays a crucial role in modulating the activity of the brainstem ANS centers. The prefrontal cortex (PFC), including the dorsolateral prefrontal cortex (DLPFC) and ventromedial prefrontal cortex (VMPFC), is associated with executive functions, decision-making, and emotion regulation. It sends top-down signals to the brainstem, including the medullary and pontine centers involved in autonomic control. The PFC can influence the balance between sympathetic and parasympathetic tone, impacting HR, blood pressure (BP), and other autonomic responses. This regulation allows the ANS to adapt to changes in the external environment and internal physiological needs. Dysregulation within these networks can contribute to the physiological and psychological symptoms of mental disorders [10].

In mental disorders such as depression and anxiety, alterations in PFC activity can disrupt the balance between sympathetic and parasympathetic activity. Dysfunctional PFC connectivity within the default mode network (DMN) and cognitive control network may be implicated in these disorders. Irregularities within the DMN, encompassing the PFC, posterior cingulate cortex, and other associated regions, have been implicated in various psychiatric disorders [14,15]. Altered connectivity within the DMN can impact self-referential thinking and emotion processing, potentially influencing ANS function. Dysregulation within the DMN is observable in conditions such as anxiety disorders, mood disorders, and schizophrenia [11,12]. In mental disorders, disruptions in these neural networks can result in altered ANS responses to emotional and physiological cues. These disruptions may lead to heightened sympathetic arousal, reduced parasympathetic activity, and an overall imbalance in ANS functioning, which can manifest as physical symptoms and impact emotion regulation. Understanding these neural network dysregulations is crucial for developing targeted interventions aimed at restoring ANS balance in individuals with mental disorders.

Impaired physiological and emotional regulation, as indicated by a reduced resting HRV, has been linked to heightened physiological and emotional responses when exposed to stress [16,17]. Reduced HRV, often indicative of ANS dysfunction, is linked to various psychiatric conditions, including anxiety, depression, schizophrenia, and post-traumatic stress disorder [18]. This decrease in HRV appears to signify a reduction in cardiac vagal tone and an increase in sympathetic activity among individuals experiencing anxiety and depression. In the context of healthy adults, HRV demonstrates a correlation with positive mood, and this association is influenced by the routine implementation of cognitive–emotion regulation strategies [19]. Increased HRV serves as a defensive barrier, attenuating the negative impact of stressors [20]. Conversely, individuals with depression and anxiety demonstrate reduced HRV when compared to those without these conditions [21,22]. Additionally, women demonstrate an elevated baseline HRV compared to men, manifested by increased power in the HF band, indicating enhanced parasympathetic activity and improved emotion regulation [23]. Conversely, individuals with schizophrenia exhibit a heightened sympathetic nervous system and parasympathetic nervous system activity, reflected in reduced HRV parameters. HF HRV was significantly reduced in patients with schizophrenia relative to healthy controls [24,25,26]. Furthermore, recent research indicates a correlation between diminished HRV and heightened severity of both positive and negative symptoms in schizophrenia. Investigations have delved into the prospect of utilizing HRV as a potential biomarker for schizophrenia [27].

In summary, abnormalities in HRV exhibit a close association with the neural networks operating within the ANS. Dysfunction in neural control, arising from both central and peripheral components, has the potential to perturb the delicate equilibrium between sympathetic and parasympathetic activity. This disruption results in discernible alterations in HRV, as evidenced in conditions such as PLMS and various psychiatric disorders. The assessment of HRV and a comprehensive understanding of its neural foundation yield valuable clinical insights. These insights may serve as a basis for interventions aimed at reinstating autonomic balance, with the ultimate goal of enhancing overall health outcomes.

## 3. ANS Dysregulation and PLMS

Although HRV reflects the neurogenic regulation of HR, lower parameters of HRV analysis have indicated a reduction in autonomic regulation. Therefore, should we expect altered HRV in PLMS, which is associated with ANS dysfunction? Information from the existing literature highlights a reduction in HRV in individuals with PLMS, coupled with a rise in sympathetic tone leading to transient autonomic disturbances during sleep [28,29,30].

PLMS can increase daytime sleepiness and subjective sleep disturbances in patients with obstructive sleep apnea due to heightened sympathetic activation [31]. Individuals with insomnia often exhibit signs of autonomic dysfunction, particularly with diminished vagal activity. Yet, practicing slow, controlled breathing can help boost vagal function and enhance sleep quality [31].

Disturbances in HR and BP signify alterations in ANS function in the pathophysiology of PLMS. This heightened sympathetic activity has the potential to disrupt the architecture of sleep, thereby contributing to sleep-related symptoms. Barone et al. conducted a study that identified changes in HRV in PLMS patients, revealing a significant reduction in HF HRV and an elevation of very low-frequency HRV [32]. Previous studies have also reported an increase in very low-frequency HRV, LF HRV, and the LF/HF ratio during PLMS events [27,29,30,33,34]. In a specific investigation [35], an elevation in very-low-frequency HRV and LF HRV was observed several tens of seconds before the onset of the PLMS period, followed by a subsequent decrease in HF HRV fluctuation. These results would, therefore, indicate a partial withdrawal of intracardiac parasympathetic activity, most likely in favor of sympathetic activity.

Traditionally, the sympathetic and parasympathetic branches of the ANS are considered to have opposing actions: sympathetic activation accelerates HR, while parasympathetic activation tends to decelerate it. This balance between the two branches is crucial for maintaining cardiovascular stability. This does not mean, however, that the heart cannot be simultaneously stimulated with sympathetic and parasympathetic activity, which may lead to the so-called “autonomic conflict” [36]. In our last study, we observed that following eight PLMS in a series, HF HRV was increased with no decrease in HR and BP [37]. The increased HF HRV suggested an increase in intracardiac parasympathetic activity, but the lack of change in HR and BP indicated a parallel, antagonistic SNS effect on the heart. Such a coactivation proposes a scenario where sympathetic and parasympathetic activities increase simultaneously, challenging the conventional notion of a reciprocal relationship. Such a phenomenon may have implications for our understanding of autonomic regulation, especially in specific contexts such as the occurrence of successive PLMS. Coactivation of the sympathetic and parasympathetic systems might indicate a more intricate regulatory mechanism, potentially reflecting a dynamic adaptation to specific pathophysiological demands. The precise mechanisms driving this coactivation in PLMS and its implications for cardiovascular health and overall physiological homeostasis will likely be subjects of further investigation.

As indicated above, psychiatric disorders may be an independent cause of ANS dysfunction, as reflected by an altered HRV. However, can ANS dysfunction constitute the basis for PLMS and thus be a bridge connecting mental dysfunctions with PLMS? Numerous studies have demonstrated that individuals with depression and anxiety disorders exhibit reduced HRV compared to healthy controls. This diminished HRV is often associated with increased sympathetic activity, which may exacerbate sleep disturbances, including PLMS. Conversely, the sleep disruption caused by PLMS can further exacerbate mood disorders, creating a vicious cycle. There is also evidence of shared neurobiological pathways between ANS function, sleep regulation, and mood regulation. Dysfunctions in neurotransmitter systems such as serotonin, norepinephrine, and dopamine can affect both ANS activity and mood regulation [38]. Moreover, chronic ANS dysregulation, along with sleep disruption caused by PLMS, may contribute to systemic inflammation. Inflammation is increasingly recognized as a contributing factor in the development of psychiatric disorders, particularly mood disorders like depression [39]. Moreover, addressing ANS dysregulation and sleep disturbances caused by PLMS through lifestyle modifications, relaxation techniques, and medications (e.g., dopamine agonists for PLMS) may indirectly improve psychiatric symptoms.

## 4. PLMS and Psychiatric Disorders

Finally, we can close the proposed cause-and-effect circle by describing the impact of PLMS on psychiatric disorders. Various psychiatric disorders, such as anxiety, depression, attention-deficit hyperactivity disorder (ADHD), and schizophrenia, have a higher prevalence of RLS, PLMS, or both [40,41,42]. It is highly probable that increased severity of RLS is closely associated with mood symptoms, including the recurring nature and diagnostic characteristics of major depressive disorders [43]. The interplay between the severity of psychiatric symptoms and PLMS represents a complex and multifaceted domain of investigation within the medical literature. Emerging evidence suggests a complex relationship between PLMS and psychiatric disorders, particularly anxiety and depression [44]. The presence of PLMS-related sleep disruptions may contribute to the development or exacerbation of anxiety symptoms [45]. Similarly, an increased risk of depression has been observed in individuals with PLMS [45]. The disrupted sleep patterns and associated daytime impairment may contribute to the onset or persistence of depressive symptoms.

The relationship between PLMS and anxiety/depression appears to be bidirectional. While PLMS-related sleep disturbances can worsen psychiatric symptoms, the presence of anxiety or depression can also lead to sleep disruption, creating a cycle of mutual influence [46,47]. Anxiety-related symptoms, such as heightened arousal and stress, may contribute to the manifestation of PLMS and disrupted sleep, and individuals with PLMS may be at a heightened risk of developing anxiety disorders [48]. Sleep disturbances, daytime fatigue, and physiological changes associated with PLMS could contribute to anxiety symptoms [49].

Some studies suggest an increased prevalence of sleep-related movement disorders, including PLMS, in individuals with schizophrenia [40]. However, the specific relationship between the severity of schizophrenia symptoms and the occurrence of PLMS is not yet fully elucidated. Medications commonly used in the treatment of schizophrenia, such as antipsychotics, can have effects on sleep architecture and may influence the occurrence of PLMS [50]. The relationship between medication use, symptom severity, and sleep disturbances is complex. Moreover, some studies suggest a link between ADHD severity and RLS symptoms, and more severe ADHD symptomatology was associated with RLS [42]. In general, sleep disturbances, including PLMS, can exacerbate symptoms of psychiatric disorders and negatively impact the overall well-being of individuals. Poor sleep quality may contribute to the severity of psychiatric symptoms and impact daily functioning [49].

## 5. ANS Dysregulation and Interoception

Interoception refers to the perception and awareness of the internal state of the body, including sensations related to heartbeat, respiration, and other physiological processes [51]. Dysregulation of the ANS may be related to disturbances in the activity of individual branches of the ANS, as well as imbalances between them. Anxiety disorders and depression may be associated with impaired functioning of the ANS and thus change an individual’s ability to perceive internal sensations. The correlation between HRV and interoception finds its primary basis in the influence of the ANS on both phenomena. HRV serves as an indicator of the dynamic equilibrium between the sympathetic (fight-or-flight) and parasympathetic (rest-and-digest) branches of the ANS [52]. An elevated HRV suggests a more adaptable and responsive ANS, which is intricately linked to heightened interoceptive awareness. Researchers have explored the relationship between HRV and interoception in various clinical and psychological contexts [53]. Understanding this association can provide insights into how ANS regulation influences our perception of internal bodily states and may have implications for mental health and the management of conditions related to autonomic dysfunction.

The brain plays a vital role in interoception, encompassing the processing of signals associated with HR. Regions like the insula and anterior cingulate cortex are crucial in integrating and interpreting interoceptive signals, including those originating from the cardiovascular system [54]. In initial studies, it was observed that individuals who perceive heartbeats differently exhibit distinct heartbeat-evoked potential (HEP) waves (i.e., HEPs). HEP is measured in relation to the R-wave in an electrocardiogram (ECG) test and is thought to reflect the brain’s processing of cardiac activity during behavior in the brain during an electroencephalogram (EEG) [55].

HEP refers to a specific brain response or electrical signal elicited by the heartbeat. This phenomenon is associated with interoception. EEGs are employed to record the brain’s electrical activity, enabling researchers to measure HEPs with a high level of precision [56]. HEPs are connected to mental disorders through their role in interoception [57,58]. Interoception plays a crucial role in our emotional and physiological self-awareness. Therefore, abnormalities or dysregulations in HEPs can be associated with various mental disorders, such as anxiety disorders, depression, or somatization [44,45]. Additionally, some studies have explored the potential of biofeedback interventions, incorporating techniques like controlled breathing, relaxation exercises, and meditation to enhance HRV [59]. However, at present, the available data do not conclusively establish whether paced respiration or subjective relaxation alone is necessary or sufficient for achieving positive outcomes with HRV biofeedback.

## 6. Interoception and Psychiatric Disorders

Interoception is closely related to the connection between mind and body, which is related to the interplay of psychological and physiological processes [51,60]. Practices that promote mindfulness and body awareness, such as meditation and biofeedback, can positively impact both interoception and HRV, contributing to improved autonomic regulation and, therefore, mental health [61]. While the existing literature on this topic is limited in scope, a synthesis of these studies suggests that integrating biofeedback with relaxation and meditation strategies may lead to an augmentation of HRV and increased parasympathetic activity [62]. It is essential to acknowledge the constraints of the reviewed literature, and these limitations underscore the need for further research in this area to explore its full potential.

HR, being a fundamental component of cardiovascular function, plays a significant role in interoception [63]. Individuals can consciously or subconsciously perceive their own heartbeats, and this awareness contributes to their overall sense of bodily self-awareness [64]. Variability in cardiac interoceptive sensitivity can influence how individuals interpret and respond to physiological cues related to HR. Fluctuations in HR are frequently linked to emotional states, stress, and arousal. Being aware of these HR changes contributes to emotional awareness, which has a pivotal role in cognitive function [53]. Interoceptive attention is a component of both emotion regulation and higher cognitive functions [51]. It allows individuals to recognize and interpret the physical sensations associated with emotions. For example, when you feel anxious, you may become aware of a racing heart or shallow breathing. Recognizing these bodily sensations can be a key step in managing and regulating emotional responses. Interoceptive attention also involves higher cognitive functions. The ability to consciously focus on and interpret internal bodily sensations requires cognitive processes such as attention, self-awareness, and self-thinking about one’s own thinking. Higher cognitive functions are responsible for recognizing and making sense of these sensations, as well as using this information to make decisions or engage in cognitive processes related to emotional and physical well-being [65]. Cortical areas involved in emotion processing, such as the amygdala and VMPFC, wield influence over the autonomic response to emotional stimuli [65]. The amygdala, for instance, can elicit sympathetic responses to perceived threats, while the VMPFC can mitigate these responses through inhibitory connections with brainstem autonomic centers [66]. Cortical regions engaged in higher cognitive functions, encompassing memory, attention, and perception, indirectly modulate the ANS [65]. Cognitive processes associated with stress, vigilance, and awareness can activate or suppress autonomic responses mediated by the brainstem [67].

Altered interoceptive processes have been observed in psychiatric disorders, influencing emotional experiences and contributing to symptom severity. The cingulate cortex, particularly the anterior cingulate cortex, serves as a crucial bridge between cognitive and emotional processes and autonomic control. It possesses the ability to regulate the brainstem nuclei governing ANS functions, thereby impacting HRV and visceral responses. The insular cortex, a hub for interoception, enables the brain to monitor and interpret internal bodily states. Its intricate connections with brainstem regions allow for real-time adjustments of autonomic responses based on perceived visceral sensations [68,69]. Dysfunctions within the insular cortex can disrupt autonomic equilibrium, contributing to psychiatric disorders [69].

Understanding how interoception processes interact with psychiatric conditions offers insights into their shared pathophysiological mechanisms. In simpler terms, individuals with higher HRV tend to be more attuned to their internal bodily sensations, such as heartbeat, breathing, and digestive processes. This enhanced interoceptive awareness can have implications for emotion regulation, stress response, and overall well-being.

## 7. Interoception and PLMS

Currently, available knowledge about the direct relationship between interoception and PLMS is limited. Although there is no well-established direct link between interoception and PLMS, it is important to note that both phenomena are based on neurobiological mechanisms with the involvement of the ANS system. Sleep disorders, including those associated with PLMS, can affect the way we perceive internal body sensations. The object of the study by Sandri et al. was an assessment of interoception in people with RLS [70]. The results indicated reduced interoceptive accuracy (measured by a heartbeat tracking task) in RLS patients. Additionally, interoceptive accuracy is negatively correlated with nocturnal eating behavior [70]. The neurobiological mechanisms underlying the relationship between impaired interoception and PLMS are unknown, but considering that PLMS is associated with disturbed sleep, it can be assumed that factors related to sleep and sleep disorders may play an important role here.

Sleep disorders have the potential to impact the ANS responsible for overseeing involuntary bodily functions, such as heart rate regulation. Conditions such as insomnia may induce imbalances in autonomic activity, potentially influencing the perception of internal bodily sensations, particularly in the context of cardiac interoception. Insomnia, characterized by challenges in falling or staying asleep, is frequently linked to elevated arousal levels. This heightened state of arousal can result in an individual’s awareness of internal sensations [71]. The examination of interoceptive sensitivity involves assessing the cerebral cortical response to an individual’s heartbeat using HEP measurement. Research by Yishul Wei et al. indicated that people suffering from insomnia had an altered HEP amplitude via frontal electrodes compared to a control group [71]. Increased neural activity temporally related to heartbeats was noted in the anterior cingulate cortex/medial frontal cortex. This implies that these individuals exhibited inadequate adjustment of the brain’s reaction to regularly recurring heartbeats [71,72]. Moreover, the interconnection of stress and sleep has potential implications for the field of sleep and emotion research. Of note, the effects of sleep deprivation on ANS dysregulation may result in dysregulation characterized by changes in sensitivity and response to stress. As a result, changes in arousal levels or interoceptive signals may affect the ability to process emotionally relevant information [73]. The relationship between cardiac interoception and PLMS is unclear, and research in this area is ongoing. However, it seems that sleep disorders associated with PLMS play a potential role. The interplay between cardiovascular interoception and sleep is bidirectional, with changes in one system influencing the other.

## 8. The Complex Interplay of Physiological, Neurobiological, and Psychological Factors Involved in PLMS and Psychiatric Disorders—Summary

The vicious circle begins as psychiatric disorders and neurological conditions, such as PLMS, give rise to disturbances in both interoception and HRV. These disturbances, in turn, contribute to the exacerbation of symptoms associated with psychiatric disorders and PLMS.

This relationship can be characterized multimodally. Shared neurochemical pathways as perturbations in dopaminergic neurons have been implicated in both PLMS and mood disorders [45,74]. Dopamine, a neurotransmitter crucial for mood regulation, may exhibit dysregulation in individuals with PLMS, potentially contributing to mood-related symptoms. Notably, PLMS frequently co-occurs with RLS, which is associated with mood disorders such as depression and anxiety. Consequently, psychiatric symptomatology frequently seen in RLS patients can indirectly be linked to PLMS in cases of co-occurrence. This reduction in quality of life can intensify feelings of distress and exacerbate psychiatric symptomatology. Then, prolonged sleep disruption, as observed in PLMS, can disrupt emotion regulation and mood stability, as sleep plays a pivotal role in modulating emotion processes [73]. The cumulative effect of PLMS-induced sleep disruption, coupled with the discomfort associated with limb movements, significantly diminishes an individual’s overall quality of life. RLS can instigate a chain reaction, ultimately linking to impaired cardiac regulation and the potential development of cardiovascular disease (Figure 2).

## 9. Clinical Implications and Future Research

Understanding the intricate interplay between HRV, ANS dysregulation, PLMS, and psychiatric disorders has clinical implications. Monitoring HRV and ANS activity in individuals with PLMS and comorbid psychiatric conditions may provide insights into treatment strategies. Interventions targeting ANS regulation, such as biofeedback and relaxation techniques, may have therapeutic potential in managing PLMS-related sleep disturbances and improving mental health outcomes. While the existing literature on this topic is limited in scope, a synthesis of these studies suggests that integrating biofeedback with relaxation and meditation strategies may lead to an augmentation of HRV and increased parasympathetic activity. Overall, the limitations of past studies underscore the need for further research in this area to explore its full potential.

## 10. Conclusions

In summary, this review underscores the need for a comprehensive approach to assessing and managing individuals with comorbid PLMS and psychiatric disorders. The interplay between PLMS, ANS dysregulation, HRV abnormalities, and psychiatric disorders represents a multifaceted and evolving field of study. HRV serves as a valuable indicator of ANS activity, reflecting the balance between sympathetic and parasympathetic branches. Reduced HRV often signifies ANS dysregulation, characterized by a heightened sympathetic tone and reduced parasympathetic influence. Interoception, on the other hand, involves the awareness and perception of internal bodily states, including autonomic responses. It plays a crucial role in emotion processing and self-regulation.

The relevance of this manuscript lies in its potential to enhance the understanding of the interactions between periodic limb movements during sleep (PLMS), heart rate variability (HRV), and psychiatric disorders, offering important insights for clinical practice. The review highlights how PLMS, a condition often co-occurring with sleep disturbances and psychiatric conditions, can be associated with autonomic nervous system (ANS) dysregulation, as reflected by changes in HRV. This understanding can assist clinicians in identifying at-risk individuals for mental health conditions based on physiological markers, such as HRV, even before more obvious symptoms appear. HRV could act as a non-invasive biomarker for assessing the effects of PLMS on the ANS and psychiatric health. This opens up opportunities for clinical applications like early diagnosis; monitoring HRV in patients with sleep complaints may aid in the early detection of autonomic dysfunction and psychiatric disorders.

Furthermore, understanding the physiological impact of PLMS on HRV and the autonomic nervous system could enable more personalized therapeutic approaches, addressing both the psychological and physiological aspects of the condition. Additionally, tracking treatment efficacy through changes in HRV over time could provide a valuable metric for evaluating the effectiveness of interventions for PLMS and associated psychiatric disorders.

In light of these possibilities, further research is warranted to elucidate the underlying mechanisms and establish effective interventions that target ANS regulation. Such efforts could transform clinical paradigms, promoting a more holistic approach to mental health and sleep disorders, ultimately fostering a deeper understanding of how physiological markers can guide tailored treatments for improved patient outcomes.

## Figures and Tables

**Figure 1 jcm-13-06129-f001:**
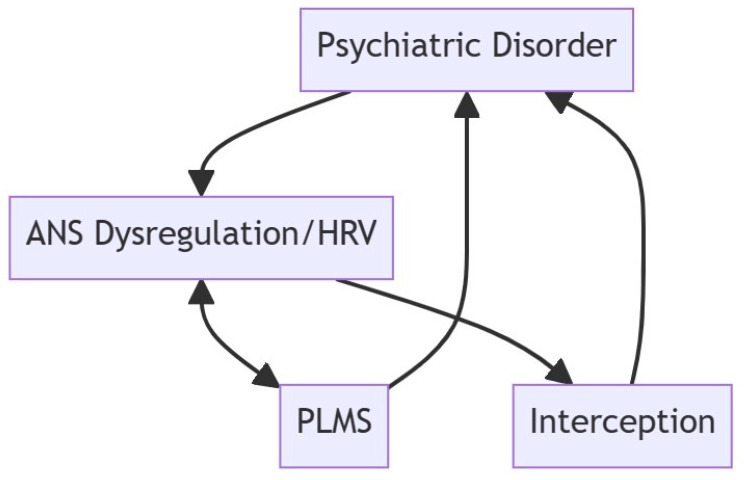
The intricate interplay between psychiatric disorders, ANS/HRV dysregulation, PLMS, and interoception; ANS—autonomic nervous system, HRV—heart rate variability, PLMS—periodic limb movements in sleep. Note: At the core of this interconnected web lies a bidirectional relationship. Psychiatric disorders contribute to ANS and HRV dysregulation, which, in turn, influence the manifestation of PLMS. Simultaneously, PLMS can exacerbate psychiatric symptoms, creating a feedback loop. Interoception—the perception of internal bodily sensations—acts as a pivotal link, influencing and being influenced by both psychiatric disorders and ANS dysregulation. This circular relationship underscores the dynamic nature of the interactions, emphasizing the mutual influence and feedback loops among these elements, illustrating the multifaceted nature of the relationship between psychiatric health, autonomic function, sleep movement disorders, and the perception of internal bodily states.

**Figure 2 jcm-13-06129-f002:**
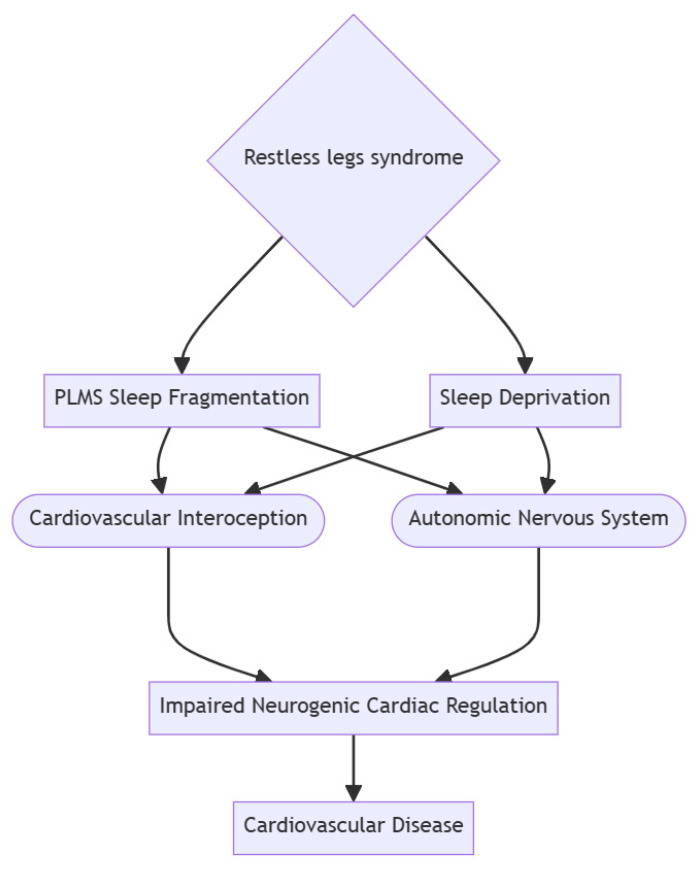
RLS initiates a chain of events leading to PLMS, sleep fragmentation, and sleep deprivation; RLS—restless leg syndrome, PLMS—periodic limb movements in sleep. Note: These events disrupt the ANS and alter interoception. The compromised ANS and impaired interoception contribute to disrupted neurogenic cardiac regulation, potentially leading to cardiovascular disease. This closed-loop sequence highlights the interdependence and sequential impact of RLS on cardiovascular health.

## Data Availability

No new data were created or analyzed in this study.

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
