# Peer review of "Heart Rate Variability and Interoception in Periodic Limb Movements in Sleep: Interference with Psychiatric Disorders?"

_jcm, 2024, doi:10.3390/jcm13206129_

Round 1
Reviewer 1 Report
Comments and Suggestions for Authors
The authors presented the review explores the complex relationships between periodic limb movements in sleep (PLMS), heart rate variability (HRV), autonomic nervous system (ANS) dysregulation, and their impact on psychiatric disorders. It's an interesting topic and the review is quite well written. However, I have some comments:
1) Authors wrote: “Interoception – the perception of internal bodily sensations – acts as a pivotal link, influencing and being influenced by both psychiatric disorders and ANS dysregulation.” I don't think Figure 1 captures the two-way relationships that the authors write about. Changes need to be made to Figure 1.
2) It is worthwhile to pay more attention to promising measures for assessing chaotic and nonlinear dynamics in HRV.
Author Response
Dear Reviewer,
Thank you very much for your thoughtful and constructive feedback on our manuscript. We appreciate your positive comments on the relevance and quality of our review.
Regarding your suggestion about Figure 1, we understand your concern about the need to clearly capture the two-way relationships between interoception, psychiatric disorders, and autonomic nervous system (ANS) dysregulation. However, we respectfully believe that the current version of the figure adequately illustrates these interactions. The arrows and structure of the diagram were designed to reflect the bidirectional nature of these relationships, though perhaps not as explicitly as you envisioned.
That being said, we would be happy to reconsider clarifying the caption or adding additional annotations to the figure to further emphasize this dynamic interaction if needed, but we do feel the current structure conveys the intended complexity of the relationships.
In response to your suggestion, we have added a relevant section discussing the potential use of these methods (yellow text). However, we would like to emphasize that the primary focus of our work is not on the methods of estimating HRV values, but rather on exploring the relationships between PLMS, HRV, autonomic nervous system dysregulation, and psychiatric disorders.
You can find the added part also here: "HRV reflects the variation in heartbeat intervals, regulated by the ANS, and is influenced by physiological factors. Studies show that while a healthy heart exhibits complex nonlinear dynamics and long-range correlations, the reduction of HRV in some medical conditions lead to reduction of its complexity by disrupting these patterns(8)(9). Inclusion of nonlinear HRV measures that evaluate data structure and organization shows promise for improving HRV interpretation. Assessing chaotic and nonlinear dynamics in HRV can be used for understanding the complex behavior of the cardiovascular system, particularly its adaptive responses to physiological and pathological conditions. While traditional time- and frequency-domain measures of HRV are useful, they often fail to capture the intricate dynamics of heart rate regulation. Nonlinear methods, inspired by chaos theory and fractal mathematics, can provide deeper insights into the complex variability of heart rate signals(10)".
Thank you once again for your insightful feedback. We look forward to hearing your thoughts.
Best regards,
Marta Małkiewicz

Reviewer 2 Report
Comments and Suggestions for Authors
Periodic limb movements during sleep are a fairly common problem associated with insomnia. The authors of the manuscript present fundamental substantiations of the relationship between periodic limb movements during sleep and mental disorders. At the same time, it has long been known that anxiety negatively affects sleep, thereby causing various types of sleep disorders, including those accompanied by periodic limb movements during sleep. In this regard, the novelty of the manuscript is not obvious. The authors also point to a change in heart rate, but there are other tests for recording autonomic nervous system dysfunction. Did the authors use these methods? Thus, the conclusions made by the authors of the manuscript do not have a clear evidence base. The authors are advised to revise the structure of the manuscript by adding such classic sections as materials and methods, discussion. The authors should also disclose the relevance of the manuscript by indicating the possibilities of clinical application of the results they obtained.
Author Response
Dear Reviewer,
Thank you for your thorough review and thoughtful comments. We appreciate your insights regarding the scope of our manuscript. Please find our response below.
“The authors also point to a change in heart rate, but there are other tests for recording autonomic nervous system dysfunction. Did the authors use these methods? “
We would like to clarify that the primary focus of our work is not on the direct analysis of autonomic nervous system (ANS) dysfunction through specific testing methods. Instead, our manuscript aims to explore the broader relationships between periodic limb movements during sleep (PLMS), heart rate variability (HRV), and psychiatric disorders, with a particular emphasis on the interplay between these factors. While we acknowledge the existence of various tests for assessing ANS dysfunction, these methods were not the central focus of our research. Therefore, we did not employ such tests in our study, as our aim was to provide a conceptual and theoretical overview rather than a detailed clinical investigation.
“The authors are advised to revise the structure of the manuscript by adding such classic sections as materials and methods, discussion”. We respectfully disagree with the reviewer’s suggestion. While incorporating sections like Materials and Methods and Discussion can enhance clarity, the nature of this manuscript as a traditional review does not necessitate a detailed methodology. A typical review does not require these classic sections, as the focus is on synthesizing existing literature rather than presenting original research.
The authors should also disclose the relevance of the manuscript by indicating the possibilities of clinical application of the results they obtained. The following text was added: “This understanding can assist clinicians in identifying at-risk individuals for mental health conditions based on physiological markers such as HRV, even before more obvious symptoms appear. HRV could act as a non-invasive biomarker for assessing the effects of PLMS on the ANS and psychiatric health. This opens up opportunities for clinical applications like early diagnosis; monitoring HRV in patients with sleep complaints may aid in the early detection of autonomic dysfunction and psychiatric disorders.
Furthermore, understanding the physiological impact of PLMS on HRV and the autonomic nervous system could enable more personalized therapeutic approaches, addressing both psychological and physiological aspects of the condition. Additionally, tracking treatment efficacy through changes in HRV over time could provide a valuable metric for evaluating the effectiveness of interventions for PLMS and associated psychiatric disorders.
In light of these possibilities, further research is warranted to elucidate the underlying mechanisms and establish effective interventions that target ANS regulation. Such efforts could transform clinical paradigms, promoting a more holistic approach to mental health and sleep disorders, ultimately fostering a deeper understanding of how physiological markers can guide tailored treatments for improved patient outcomes.”
Thank you again for your valuable feedback, which has significantly contributed to the refinement of our manuscript. We hope that these revisions meet your expectations and enhance the overall quality of our work.
With regards,
Marta Małkiewicz

Reviewer 3 Report
Comments and Suggestions for Authors
Minor Changes:
1. All figs quality should be high.
2. At least add the 150 articles in this study.
3. Title should be revise.
4. Follow some standard review to revise it.
Major Changes:
5. Author should revise the article carefully.
6. This review is very short; it is written like mini-review. Author need to address all questions.
7. Author is talking about PLMS, so first need to talk about brain, sleep, sleep disorders, and then PLMS.
Heyat, Md BB, et al. "Detection, treatment planning, and genetic predisposition of bruxism: a systematic mapping process and network visualization technique." CNS & Neurological Disorders-Drug Targets (Formerly Current Drug Targets-CNS & Neurological Disorders) 20.8 (2021): 755-775.
8. In the Introduction, author should add the main contributions of the study.
9. Author should add the Methods such as PRISMA, etc. of the data collection.
1. Author should add the tables and figures in each section. Because theoretical review is not attractive for the researchers.
. Author should add the bibliometric analysis in the study.
1. Author should need to add the discussion section.
1. Author should add the biological diagrams and the role of AI in sleep disorders.
Comments on the Quality of English LanguageMINOR
Author Response
Dear Reviewer,
Thank you for the detailed feedback, which will help improve the manuscript. However, I would like to address a few points. Firstly, the paper is sufficiently comprehensive and detailed, so it should not be considered a mini-review. It covers all the key aspects of the topic in depth. Secondly, I have chosen not to use the PRISMA method as this is a theoretical review, and PRISMA is typically used for systematic reviews, which is not the focus of this study. Thirdly, I do not see the need to change the title, as it accurately reflects the content and scope of the paper while maintaining a logical flow of the discussed topics. Finally, the structure of the manuscript is well-organized and logical. We start by addressing important topics related to PLMS and then gradually move into more detailed discussions, following a natural progression of the subject. Regarding the request to add tables and figures in each section, while I understand that visual aids can enhance the attractiveness of a paper, this is primarily a theoretical review, and the content is based on conceptual analysis. I believe the current structure, supported by key figures and tables where necessary, is sufficient to convey the intended information. However, I will consider adding more visual elements if they add substantive value to the discussion. As for the bibliometric analysis, it is not within the scope of this paper. The aim of this review is to provide a focused theoretical perspective on the subject rather than perform a bibliometric study. Therefore, I do not see the necessity of including this component. I acknowledge the suggestion to include a discussion section and it is developed further in the end, ensuring it addresses key findings and their implications.
Lastly, regarding the inclusion of biological diagrams and the role of AI in sleep disorders, I appreciate this recommendation. While AI is an interesting and evolving topic, it is not within the scope or focus of this paper. The primary objective of this review is to explore the theoretical and biological aspects of PLMS, HRV and related psychiatric disorders, rather than delve into AI applications. Therefore, I do not see the need to include discussions about AI, as it would divert from the main purpose of this study.
Thank you once again for your valuable suggestions and insights. I appreciate the time and effort put into reviewing the manuscript. We hope this explanation clarifies our approach. Below you will find the revised manuscript with marked changes.
I look forward to improving the manuscript and ensuring it remains clear, concise, and relevant to the subject matter.
With regards,
Marta Małkiewicz

Reviewer 4 Report
Comments and Suggestions for Authors
1. Will the combination of periodic limb twitching during night sleep increase daytime sleepiness and subjective sleep disturbance in patients with obstructive sleep apnea? Patients with obstructive sleep apnea who develop periodic limb twitching during sleep after Yang pressure breathing therapy have a higher tendency of sympathetic nerve activation in their basic autonomic nervous system, and therefore may have a higher risk of cardiovascular disease.
2. Whether this study excludes all sleep events and the segments that may have a lasting impact on heart rhythm, and whether sympathetic nerve activation tends to cause obstructive sleep apnea with periodic limb twitching during night sleep after continuous positive pressure breathing therapy. Patients also exist
3. Is there a relationship between periodic limb tics and sleep apnea? Different studies have shown that about 25% to 50% coexist. Because patients with severe sleep apnea have frequent awakenings and intermittent hypoxemia, it may be Induces sympathetic nerve activation, resulting in periodic limb movements.
4. Will the limb movements of periodic limb tics interfere with the therapeutic effect of the Yang pressure respirator, possibly causing sleep disruption and affecting the normal use of the respirator?
Author Response
Dear Reviewer,
We appreciate the your insightful comments. However, our manuscript specifically focuses on periodic limb movements during sleep (PLMS). The work includes tables and figures that support our findings and arguments. As noted, PLMS can exacerbate daytime sleepiness and sleep disturbances in patients with obstructive sleep apnea (OSA) due to heightened sympathetic activation and fragment citing Yang pressure breathing therapy is included. “PLMS can increase daytime sleepiness and subjective sleep disturbances in patients with obstructive sleep apnea due to heightened sympathetic activation(31). Individuals with insomnia often exhibit signs of autonomic dysfunction, particularly with diminished vagal activity. Yet, practicing slow, controlled breathing can help boost vagal function and enhance sleep quality(31).”
However, we would like to clarify that the primary focus of our manuscript is on PLMS, not OSA. Additionally, although limb movements may affect CPAP therapy adherence in OSA patients, our work centers on the implications of PLMS. We hope this explanation clarifies our approach. Below you will find the revised manuscript with marked changes.
With regards,
Marta Małkiewicz

Round 2
Reviewer 1 Report
Comments and Suggestions for Authors
I agree with the authors' response.
Reviewer 4 Report
Comments and Suggestions for Authors
The author answered my questions very well.